**Brief Investigation**

# Approximate Bayesian computation supports a high incidence of chromosomal mosaicism in blastocyst-stage human embryos

Qingya Yang (ID) ,[1] Sara A. Carioscia (ID) ,[1] Matthew Isada (ID) ,[1] Rajiv C. McCoy (ID) [1,*]

[1]Department of Biology, Johns Hopkins University, Baltimore, MD 21218, United States

*Corresponding author: Department of Biology, Johns Hopkins University, 3400 N. Charles St., Mudd 144, Baltimore, MD 21218, United States. Email: rajiv.mccoy@jhu.edu

Chromosome mis-segregation is common in human meiosis and mitosis, and the resulting aneuploidies are the leading cause of pregnancy loss. Preimplantation genetic testing for aneuploidy (PGT-A) prioritizes chromosomally normal embryos for transfer based on analysis of a biopsy of ~5 trophectoderm cells from blastocyst-stage in vitro fertilized embryos. While modern PGT-A platforms classify these biopsies as aneuploid, euploid, or mosaic (a mixture of normal and aneuploid cells), the underlying incidences of aneuploid, euploid, and mosaic embryos and the rates of meiotic and mitotic error that produced them remain largely unknown. To address this knowledge gap, we paired a method for embryo simulation with approximate Bayesian computation to infer rates of meiotic and mitotic error that explain published PGT-A data. Using simulation, we also evaluated the chromosomal status of entire embryos. For a published clinical sample, we estimated a 40% to 58% probability of meiotic error per meiosis and a 1.5% to 6.3% probability of mitotic error per mitosis, depending on assumptions about spatial organization. In addition, our analyses suggest that <1% of blastocysts are fully euploid and that many embryos possess low-level mosaic clones that are not captured during biopsy. These conclusions were relatively insensitive to misclassification of mosaic biopsies. Together, our findings imply that low-level mosaicism is a normal feature of embryogenesis and are consistent with clinical data demonstrating the developmental potential of mosaic-testing embryos. More broadly, our work helps overcome the limitations of embryo biopsies to estimate fundamental rates of chromosome mis-segregation in human development.

Keywords: aneuploidy; IVF; PGT-A; development; meiosis; mitosis

## Introduction

It is estimated that only approximately half of human conceptions survive to birth. The leading cause of human pregnancy losses—most of which occur very early in development—is abnormality in chromosome number (aneuploidy) (Boklage 1990; Macklon et al. 2002; Jarvis 2016). Chromosomes frequently mis-segregate during (predominantly maternal) meiosis, producing embryos with homogeneous forms of aneuploidy that affect all cells (Hassold and Hunt 2001; Gruhn et al. 2019). However, errors may also arise during postzygotic cell divisions, producing "mosaic" embryos with 2 or more karyotypically distinct cell lineages (McCoy 2017). The first 2 cell divisions appear especially susceptible to such mitotic errors (Currie et al. 2022).

While many forms of aneuploidy are lethal during cleavage-stage development, other aneuploidies, especially in mosaic form, can survive to the blastocyst stage and beyond (Capalbo et al. 2021; Viotti, Victor et al. 2021; McCoy et al. 2023). Considering the potential fitness consequences of aneuploidy, preimplantation genetic testing for aneuploidy (PGT-A) seeks to improve IVF outcomes by prioritizing chromosomally normal (i.e. euploid) embryos for transfer based on genetic analysis of embryo biopsies (Cimadomo et al. 2020), although its clinical efficacy is the subject of long-standing debate (Mastenbroek et al. 2007; Munne et al. 2007; Rubio et al. 2017; Verpoest et al. 2018; Munné

et al. 2019; Cornelisse et al. 2020). Current implementations of PGT-A involve sequencing or single-nucleotide polymorphism microarray-based genotyping of DNA extracted from trophectoderm biopsies of blastocyst-stage embryos, 5 to 6 days after fertilization (Vermeesch et al. 2016). These modern, sensitive PGT-A platforms have revealed evidence of mosaic aneuploidy within 2% to 26% of embryo biopsies, posing a dilemma for clinical diagnosis and management (Popovic et al. 2020).

One notable limitation of PGT-A is its dependence on a single, spatially restricted biopsy of 5 to 10 trophectoderm cells, which may or may not represent the constitution of the remaining >100 cells of the blastocyst-stage embryo, including the inner cell mass (Esfandiari et al. 2016; Vera-Rodriguez and Rubio 2017). While studies employing multiple biopsies of individual embryos (Victor, Griffin et al. 2019; Kim et al. 2022) or single-cell genomic analysis (Starostik et al. 2020; Gallardo et al. 2023) have offered additional insight, they are typically limited in sample size of embryos and/or availability of ground-truth data for comparison. Thus, the clinical literature is replete with reports of the relative incidences of euploid, mosaic, and aneuploid embryo biopsies as diagnosed by PGT-A, but the underlying incidence of mosaic embryos and the fundamental rates of meiotic and mitotic errors that produced them remain largely obscure.

Approximate Bayesian computation (ABC) offers a statistical framework for estimating unknown parameters given summary

statistics from observed data and a generative model for simulating data under a specified set of parameters (Sunnaker et al. 2013). ABC is most useful in situations such as ours where the likelihood function is complex or intractable, but efficient simulation is possible. In addition to point estimates of the parameters, the posterior distributions produced by ABC provide measurements of uncertainty that can be summarized, for example, with credible intervals (CIs). ABC has found growing applications in the fields of population genetics, ecology, and epidemiology, in concert with the development of powerful computational simulations (Beaumont 2010).

Skinner et al. (2024) recently developed an R package, Tessera, to model the growth and distribution of aneuploid cells in a 3D early embryo, followed by a trophectoderm biopsy to mimic PGT-A. In our study, we used Tessera to simulate embryos over a range of meiotic and mitotic error rates. We then applied ABC to identify the subset of embryos whose biopsy results best reflect published PGT-A data from IVF clinics, in turn obtaining posterior probability estimates of meiotic and mitotic error rates and incidences of euploid, mosaic, and aneuploid embryos. Together, our study helps overcome the limited focus on embryo biopsies to reveal knowledge of fundamental parameters shaping the chromosomal landscape of human preimplantation embryos.

## Results

### ABC enables inference of rates of meiotic and mitotic error

We used Tessera to simulate large samples of human embryos and biopsies within the ABC framework (see Methods; Fig. 1) and compare to published data from PGT-A. For simplicity, we focused our analysis on published reports of aneuploidies affecting whole chromosomes, as segmental aneuploidies (duplication or deletion of subchromosomal segments) originate from distinct mechanisms, and criteria for their reporting vary substantially across clinics (Girardi et al. 2020).

Given a rate of meiotic error per meiosis (i.e. the probability of producing an aneuploid zygote, ignoring the distinction between maternal and paternal meiosis) drawn from a uniform prior distribution, a rate of mitotic error per mitosis (i.e. the probability of a postzygotic cell division producing aneuploid daughter cells) drawn from a uniform prior distribution, and a rate of dispersal (i.e. the spatial organization of aneuploid cells within mosaic embryos) fixed at 0, 0.5, or 1, we simulated trials of 1,000 embryos each. By definition, a dispersal rate of 0 denotes complete spatial clustering, whereby aneuploid cells are adjacent to one another, a dispersal rate of 1 denotes maximum spatial separation of aneuploid cells, and intermediate values represent more moderate patterns of spatial clustering. We selected dispersal values of 0, 0.5, and 1 to span the entire parameter range, as the spatial distribution of aneuploid cells within mosaic embryos remains largely unknown (see Discussion).

We then simulated biopsies of 5 spatially adjacent cells from each embryo and summarized each trial based on the proportion of euploid, mosaic, and aneuploid biopsies. A biopsy was categorized as mosaic if any but not all of the 5 sampled cells were aneuploid. Within the ABC framework, we compared these proportions to those from published PGT-A results from 6,766 clinical trophectoderm biopsies (23% euploid, 19% mosaic, and 58% aneuploid) by Capalbo et al. (2021) and rejected simulations whose distance from the published data exceeded a given tolerance threshold. For the remaining selected simulations, we used the

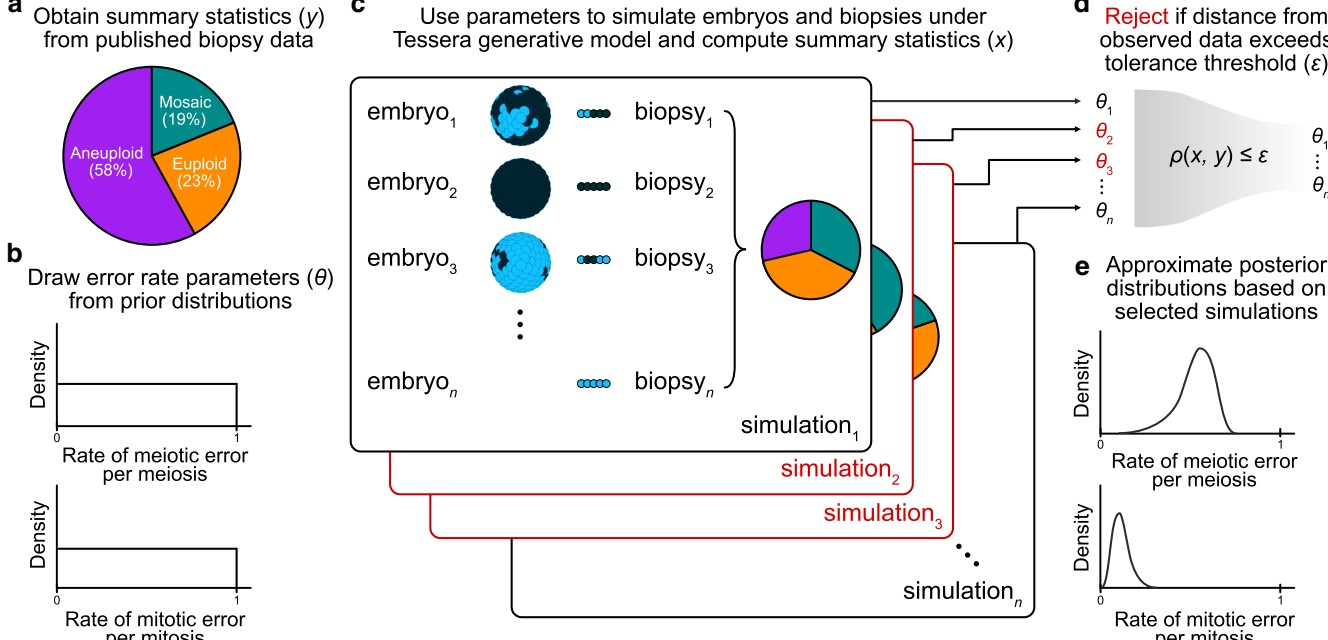

**Fig. 1.** Schematic of the ABC approach for inferring meiotic and mitotic error rates. a) Observed proportions of euploid, aneuploid, and mosaic biopsies from PGT-A were obtained from the literature (Capalbo et al. 2021) and used as summary statistics for ABC. b) For each simulation trial, probabilities of meiotic and mitotic error (θ) were randomly drawn from uniform prior distributions ranging from 0 to 1. c) These values were used to determine the number of cells that would be aneuploid (represented as dark blue in the schematic) vs normal disomic (light blue) in the final embryo. This number was converted to a proportion and then used as input to the Tessera package. For each combination of error parameters and each of 3 fixed values for dispersal (0, 0.5, and 1), 1,000 embryos were generated and biopsied. For each embryo, 8 rounds of mitotic division were simulated (producing a total of 256 cells), and one 5-cell biopsy was obtained. d) Using the adaptive method described by Lenormand et al. (2013), ABC selected the set of simulations that produced summary statistics (proportions of euploid, mosaic, and aneuploid biopsies) that best reflected the observed biopsy data. e) The error rates that produced the selected simulations were used to approximate the posterior distributions of these parameters.

corresponding values of the meiotic and mitotic error rate parameters to approximate posterior probability distributions. The sequence of tolerance levels, stopping criteria, and bias correction was obtained using the adaptive method described by Lenormand et al. (2013) and implemented with the EasyABC package in R (Jabot et al. 2013).

At a dispersal level of 0, the posterior mean estimates of the meiotic and mitotic error rates were 40% (95% CI [34%, 45%]) and 6.3% (95% CI [5.4%, 7.1%]), respectively; at a dispersal level of 0.5, the posterior mean estimates of the meiotic and mitotic error rates were 57% (95% CI [54%, 60%]) and 2.1% (95% CI [1.8%, 2.3%]), respectively, and at a dispersal level of 1, the posterior mean estimates of the meiotic and mitotic error rates were 58% (95% CI [55%, 60%]) and 1.5% (95% CI [1.4%, 1.7%]), respectively. These estimates are summarized in Fig. 2 and Supplementary Table 1 and indicate that higher dispersal levels (i.e. less spatial clustering of aneuploid cells) require lower estimates of mitotic aneuploidy rates and modestly higher estimates of meiotic aneuploidy rates to explain the observed data. This result is intuitive, as higher dispersal levels make it more likely to detect mosaic aneuploidy within a given biopsy, such that relatively lower mitotic error rates would be required to explain the observed data.

For the selected simulations, we observed a negative correlation between the inferred meiotic and mitotic error rates at each dispersal level (Fig. 3). The correlation was much more pronounced when the dispersal level was zero, reflecting the fact that spatial clustering of cells makes it more likely that a mitotic aneuploidy could mimic a meiotic aneuploidy (or euploid embryo) by affecting all cells (or no cells) within a biopsy. Higher assumed rates of dispersal make these forms of aneuploidy more distinguishable in biopsy data, reducing the correlation.

## Evidence that few embryos are fully euploid

We next used simulations to examine the cellular content of embryos generated under the posterior distributions of meiotic and mitotic error rates inferred with ABC from Capalbo et al. (2021) (Supplementary Table 2). To this end, we simulated new samples of embryos by drawing meiotic and mitotic error rates from the posterior distributions (at each of the 3 assumed levels of dispersal) and examined their entire cellular content. In contrast to the proportions of euploid, aneuploid, and mosaic biopsies, this allowed us to quantify the underlying proportions of fully euploid, fully aneuploid, and mosaic embryos (and their corresponding proportions of aneuploid cells) that produced these data—quantities that remain largely unknown within the field. An embryo was considered mosaic if any but not all cells within it were aneuploid, but the entire distribution of aneuploid cell proportions was also visualized.

Strikingly, we found that regardless of assumptions about dispersal, fewer than 1% of embryos from the posterior predictive samples were entirely euploid (Fig. 4a; Supplementary Table 2). Specifically, for simulated dispersal levels of 0, 0.5, and 1, we inferred that 0, $1.5 \times 10^{-5}$, and $1.6 \times 10^{-4}$ proportions of embryos were fully euploid, respectively. A large proportion of simulated embryos exhibited mosaicism, especially low-level mosaicism affecting <25% of cells when allowing for nonzero dispersal. Higher levels of dispersal were consistent with lower levels of aneuploidy within mosaic embryos and higher proportions of fully aneuploid embryos (Fig. 4a).

## Evaluating the reliability and consistency of embryo biopsies

By simulating biopsies from these posterior predictive samples, we next evaluated the extent to which embryo biopsies

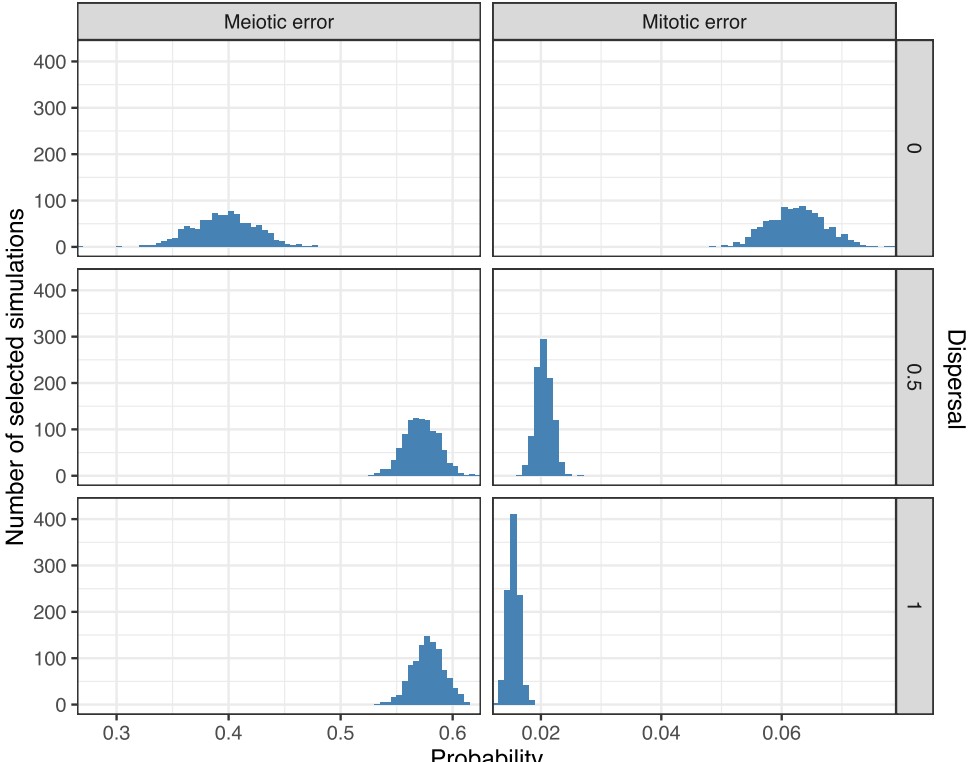

**Fig. 2.** Posterior probabilities of meiotic and mitotic error. Posterior distributions of meiotic and mitotic error probabilities (per meiosis and mitosis, respectively) based on 3,000 simulations (1,000 simulations per level of dispersal) selected by ABC as best matching published data.

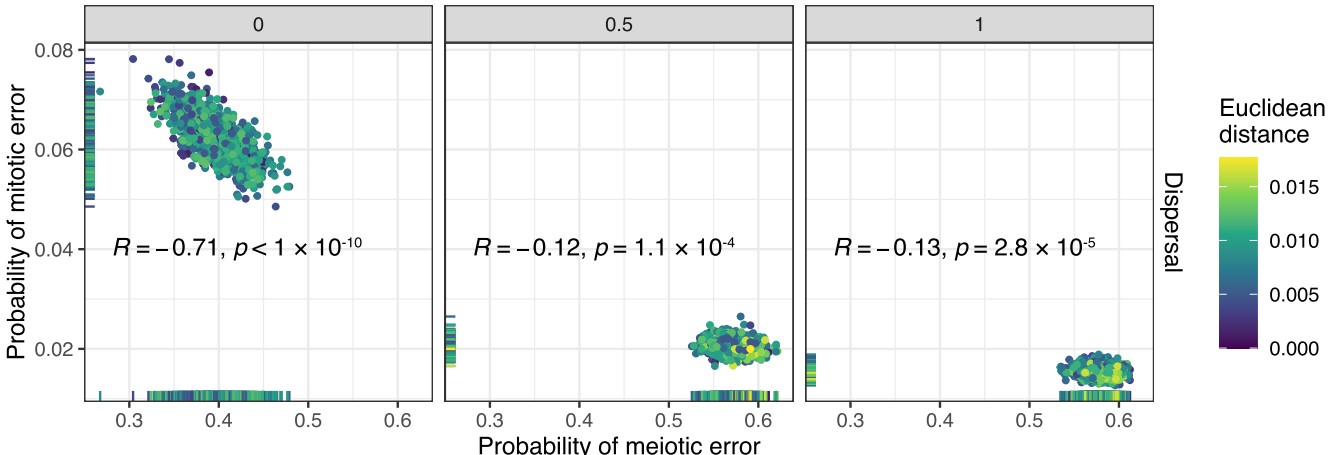

**Fig. 3.** Relationship between inferred meiotic and mitotic error rates. Meiotic vs mitotic error rate, stratified by dispersal level, across 3,000 simulations selected by ABC. The coefficient (R) and P-values of Pearson's correlations between inferred meiotic and mitotic error rates are displayed in each panel. The Euclidean distances between the vectors of summary statistics of selected simulations and the target summary statistics from the literature are indicated with colors.

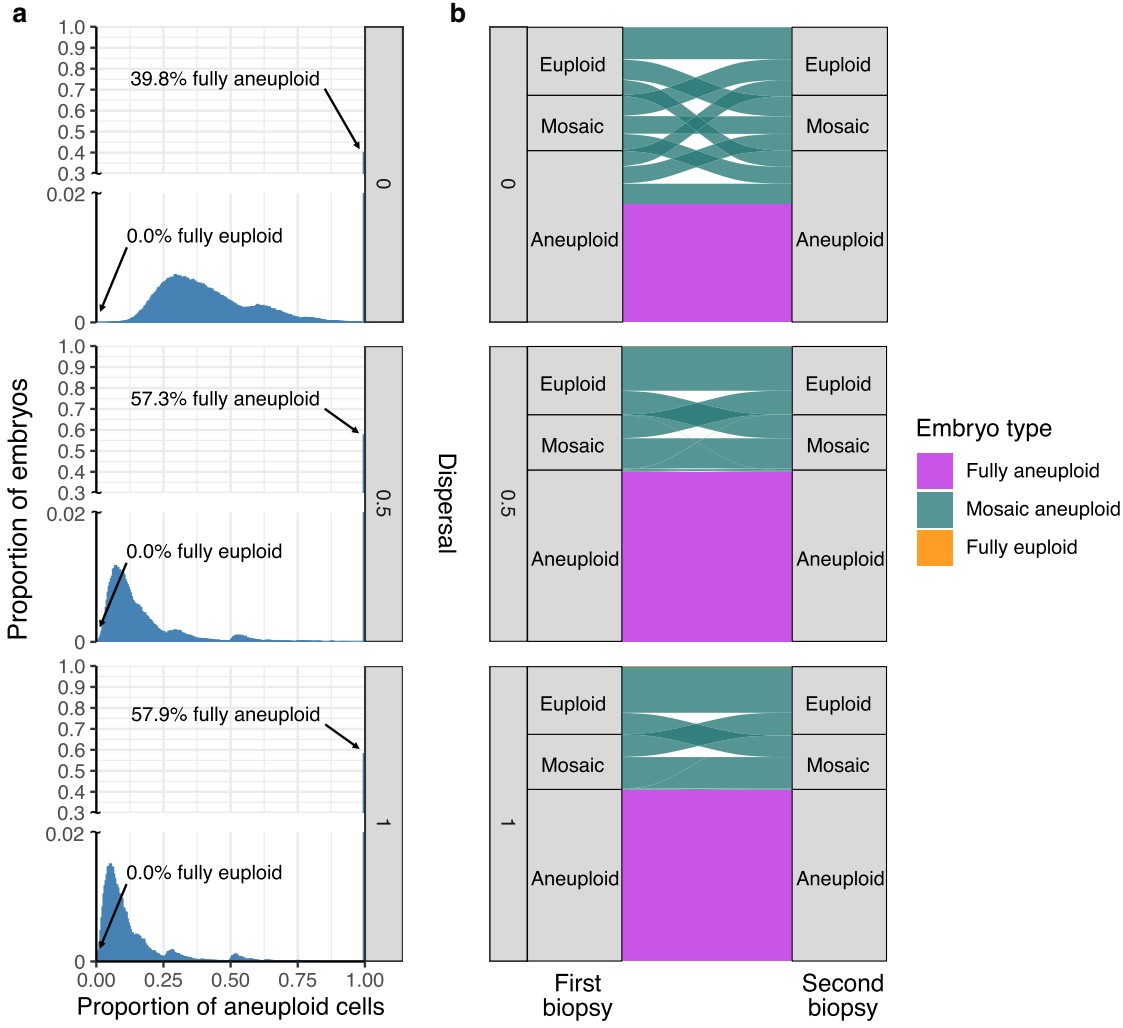

**Fig. 4.** Posterior predictive samples of embryos that are fully aneuploid, mosaic aneuploid, and fully euploid. a) Distribution of proportions of aneuploid cells in embryos simulated based on meiotic and mitotic error rates drawn from the posterior distributions (1,000,000 embryos for each level of dispersal; Fig. 2). The arrow marks the percentage of fully euploid embryos (0 aneuploid cells) in the posterior predictive sample for each dispersal level (all 0.0%). b) Comparisons between ploidy statuses of first and second biopsies from embryos of different underlying types (indicated by ribbon colors), stratified by simulated levels of dispersal. Ribbons connecting the same vs different biopsy categories indicate concordant and discordant results, respectively, while the widths of the ribbons indicate proportions.

represented the ploidy statuses of the embryos from which they originated. Intuitively, due to the small number of cells in biopsies, aneuploid cells of low-level mosaic embryos were frequently unsampled, and their embryo biopsies were more often classified as euploid than mosaic (Fig. 4b). Within a mosaic embryo, higher levels of dispersal increased sensitivity for capturing 1 or more aneuploid cells within a biopsy. This effect is apparent from the skewness of the distribution of the proportion of aneuploid cells within mosaic embryos, where higher simulated levels of dispersal required smaller proportions of aneuploid cells within mosaic embryos to explain the observed data (Fig. 4a).

Given reports of healthy births following the transfer of embryos categorized as mosaic (Greco et al. 2015; Capalbo et al. 2021; Viotti, Victor et al. 2021), an important clinical concern regards the possibility that mosaic embryos could produce a biopsy categorized as aneuploid, such that a potentially viable embryo would not be transferred. Within our posterior predictive samples, we observed that 31.5%, 1.4%, and 0.7% of aneuploid biopsies originated from mosaic embryos at assumed dispersal levels of 0, 0.5, and 1, respectively (Fig. 4b; Supplementary Table 3). This result implies that accurate interpretation of aneuploid embryo biopsies strongly depends on the assumed spatial organization of aneuploid cells within mosaic embryos, consistent with previous modeling using Tessera (Skinner et al. 2024).

In addition to cell tracking and other microscopy-based approaches, information about such spatial organization can be derived from previous studies that compared PGT-A results between multiple biopsies of individual blastocyst-stage embryos (Victor, Griffin et al. 2019; Navratil et al. 2020; Marin et al. 2021; Kim et al. 2022; Cascante et al. 2023). To mimic this study design, we simulated a second biopsy from each embryo in our posterior predictive samples and compared their results to the original biopsy (Fig. 4b; Supplementary Table 4). Across all dispersal levels, initial mosaic biopsies were often discordant with the second biopsy (31.7%, 54.5%, and 57.2% concordance at dispersal levels of 0, 0.5, and 1, respectively), consistent with the literature. However, discordance between the first and second biopsy was much more pronounced at the lowest dispersal level (dispersal = 0), whereas nonzero dispersal levels exhibited higher concordance —especially for the aneuploid class, which is of greatest clinical relevance (80.7%, 98.9%, and 99.5% concordance at dispersal levels of 0, 0.5, and 1, respectively).

## Inferences are robust to moderate rates of misclassification

In various perspective articles, authors have argued that mosaic biopsies may be over-diagnosed, as technical variability in depth of coverage (e.g. due to amplification artifacts) may be falsely interpreted as evidence of mosaicism (Capalbo et al. 2017; Treff and Marin 2021). To address the possibility of biopsy misclassification and assess its impact on our results, we adjusted the reference data from Capalbo et al. (2021) by evenly reassigning a varying proportion of mosaic biopsies as either euploid or aneuploid (half to each category; Supplementary Fig. 1) and then applied the same ABC process to this adjusted reference data. We then sampled error rate pairs from each of the newly inferred posterior distributions and generated embryos for downstream analysis. While the inferred incidence of mosaic embryos quantitatively declined with higher rates of misclassification, qualitative conclusions about the incidence of mosaic embryos were relatively insensitive to the misdiagnosis of embryo biopsies (Fig. 5). Notably, even when assuming a misdiagnosis rate as high as 90%, fewer than 10% of embryos were inferred as fully euploid, regardless of the assumed level of dispersal.

## Inferences are robust to variation across PGT-A datasets

Given that patient cohorts, laboratory procedures, and details of data analysis and reporting may vary across published PGT-A studies, we extended our approach to 3 additional input datasets (Munné et al. 2017; Rodrigo et al. 2020; Clarke et al. 2023) to evaluate the robustness of our conclusions (Table 1). The results were qualitatively consistent across studies. While meiotic error rates vary between studies (e.g. due to differences in the maternal age distribution; Supplementary Table 5), inferred rates of mitotic error remained relatively constant, and small proportions of embryos (<1%) simulated under these parameters were fully euploid.

## Discussion

Aneuploidies are prevalent in human development and are the leading cause of pregnancy loss (Hassold et al. 2007). Retrospective studies of PGT-A data typically report proportions of embryo biopsies classified as euploid, aneuploid, or mosaic. While potentially justified from a clinical perspective (as these are the classifications used to inform embryo prioritization), it is important to consider from a fundamental biological perspective that these results are based on a biopsy of ∼5 spatially restricted trophectoderm cells from a blastocyst-stage embryo composed of >100 cells. The underlying incidences of fully euploid, fully aneuploid, and mosaic embryos remain largely unknown and contentious (Capalbo et al. 2017; Vera-Rodriguez and Rubio 2017; Treff and Marin 2021), and the fundamental rates of meiotic and mitotic chromosome mis-segregation that produce these patterns have not been estimated aside from live imaging studies of the first 2 postzygotic cell divisions (Currie et al. 2022). Here, we used the statistical method ABC to address these limitations and achieve embryo-wide estimates of aneuploidy that best explain published clinical data.

Our most provocative finding is that the inferred proportion of mosaic embryos is much higher than the observed proportion of mosaic biopsies and that very few embryos are fully euploid at the blastocyst-stage of development. This conclusion, as well as the results of our re-biopsy simulation, is consistent with clinical studies performing multiple biopsies of individual embryos, which observed that "fully aneuploid" biopsies exhibited high concordance upon re-biopsy, suggesting their likely meiotic origins. Meanwhile, mosaic aneuploid biopsies exhibited low concordance, suggesting that they either originated from technical artifacts or reflected the limited, localized sampling of aneuploid clones within patchy mosaic embryos (Victor, Griffin et al. 2019; Navratil et al. 2020; Marin et al. 2021; Kim et al. 2022; Cascante et al. 2023). Recent studies leveraging single-cell sequencing data also support the conclusion that low-level mosaicism is widespread at the blastocyst stage, although they were limited in sample size of embryos and ground-truth data for validation (Starostik et al. 2020; Gallardo et al. 2023; Chavli et al. 2024; Zhai et al. 2024).

As with all models, our results must be interpreted in light of several simplifying assumptions. One such assumption regards the spatial distribution of aneuploid cells within mosaic embryos, which remains largely unknown. Past work in mouse models of experimentally induced mosaicism reported an even distribution of aneuploid cells throughout postimplantation conceptuses (day 13.5 of embryonic development), with no significant enrichment

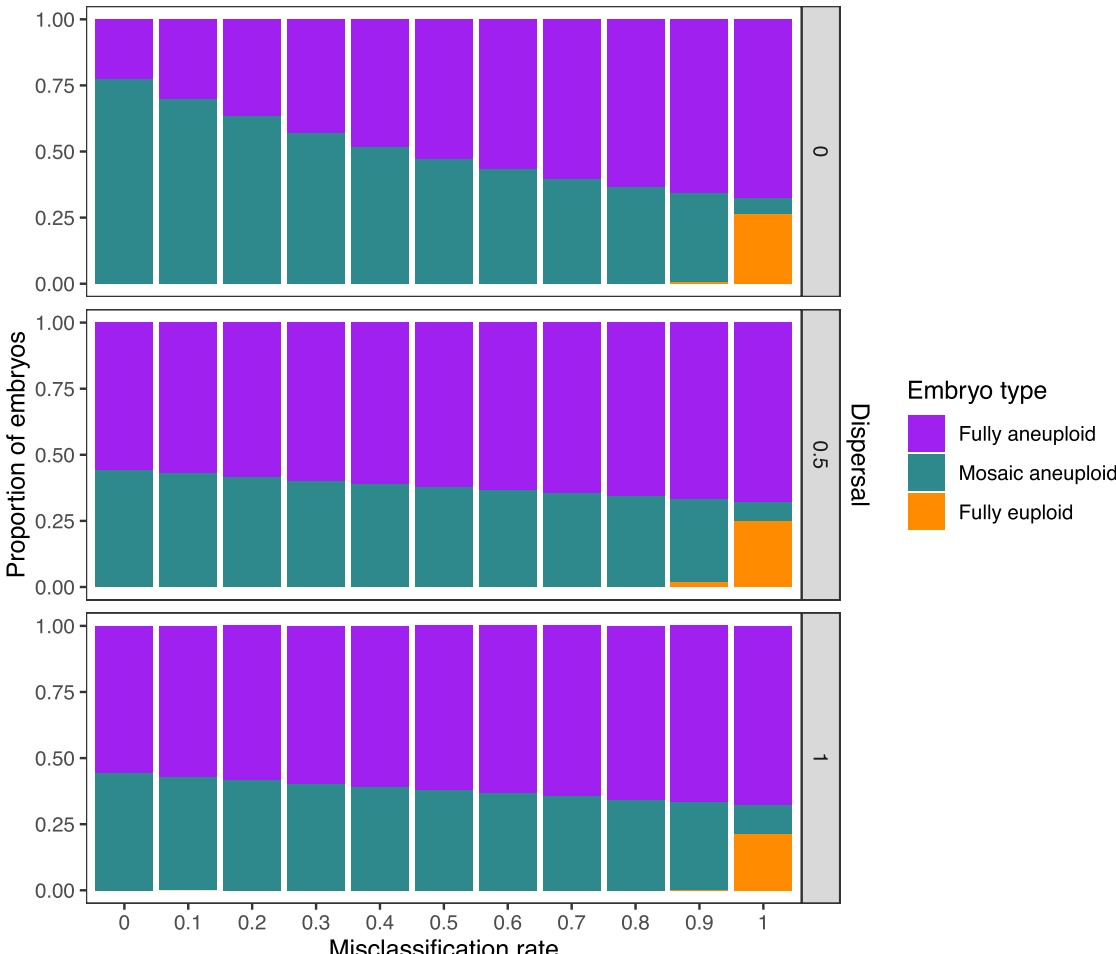

**Fig. 5.** Inferred proportions of embryo types assuming varying levels of biopsy misclassification. 1,000 pairs of error rates were sampled from the posterior distributions generated by ABC with adjusted target values (based on the rate of mosaic biopsy misclassification; Supplementary Fig. 1). Each error rate pair was used to generate 1,000 embryos, and the proportions of each embryo type were plotted. The x axis represents the proportion of published mosaic biopsies that were assumed to be false positives and evenly reassigned to euploid and aneuploid categories.

**Table 1.** Posterior mean estimates of meiotic and mitotic error probabilities inferred with ABC across a range of target proportions of euploid, mosaic, and aneuploid embryo biopsies reported in the literature based on PGT-A. Estimates are shown across 3 levels of dispersal (Disp.; 0, 0.5, and 1). The posterior distributions of meiotic and mitotic error probabilities were used to simulate embryos, and the proportions of fully aneuploid, mosaic aneuploid, and fully euploid simulated embryos are also reported.

| Dataset | Disp. | Published biopsy data | | | Inferred probabilities | | Inferred embryo types | | |
|---|---|---|---|---|---|---|---|---|---|
| | | Euploid | Mosaic | Aneuploid | Meiotic error | Mitotic error | Fully euploid | Mosaic aneuploid | Fully aneuploid |
| Capalbo et al. (2021) | 0 | 0.23 | 0.19 | 0.58 | 0.39 | 0.064 | 0.00 | 0.60 | 0.40 |
| Clarke et al. (2023) | 0 | 0.49 | 0.18 | 0.33 | 0.30 | 0.024 | 0.00 | 0.70 | 0.30 |
| Munné et al. (2017) | 0 | 0.53 | 0.15 | 0.32 | 0.31 | 0.017 | 0.00 | 0.69 | 0.31 |
| Rodrigo et al. (2020) | 0 | 0.51 | 0.06 | 0.43 | 0.38 | 0.016 | 0.00 | 0.62 | 0.38 |
| Capalbo et al. (2021) | 0.5 | 0.23 | 0.19 | 0.58 | 0.59 | 0.021 | 0.00 | 0.43 | 0.57 |
| Clarke et al. (2023) | 0.5 | 0.49 | 0.18 | 0.33 | 0.34 | 0.011 | 0.00 | 0.67 | 0.33 |
| Munné et al. (2017) | 0.5 | 0.53 | 0.15 | 0.32 | 0.33 | 0.009 | 0.01 | 0.67 | 0.32 |
| Rodrigo et al. (2020) | 0.5 | 0.51 | 0.06 | 0.43 | 0.43 | 0.011 | 0.00 | 0.58 | 0.42 |
| Capalbo et al. (2021) | 1 | 0.23 | 0.19 | 0.58 | 0.58 | 0.015 | 0.00 | 0.42 | 0.58 |
| Clarke et al. (2023) | 1 | 0.49 | 0.18 | 0.33 | 0.32 | 0.008 | 0.01 | 0.66 | 0.33 |
| Munné et al. (2017) | 1 | 0.53 | 0.15 | 0.32 | 0.32 | 0.007 | 0.02 | 0.65 | 0.33 |
| Rodrigo et al. (2020) | 1 | 0.51 | 0.06 | 0.43 | 0.42 | 0.017 | 0.00 | 0.58 | 0.42 |

in the placenta vs the fetus (Bolton et al. 2016). To address this uncertainty, we performed simulations at varying dispersal levels ranging from complete spatial clustering (dispersal = 0) to maximum spatial separation (dispersal = 1) of aneuploid cells. Importantly, our major qualitative conclusions about the

incidence of mosaic aneuploidy and underlying rates of meiotic and mitotic error were relatively insensitive to assumptions about dispersal levels. Meanwhile, the relationship between the ploidy statuses of embryo biopsies and underlying embryos, as well as the concordance between multiple embryo biopsies, was more

sensitive to assumed dispersal, consistent with previous work (Skinner et al. 2024). Specifically, the dispersal level of zero exhibited the highest rate of discordance, including a substantial risk that mosaic embryos will produce aneuploid biopsies. Given that published studies have typically observed that embryos producing aneuploid biopsies exhibit high concordance upon re-biopsy (Victor, Griffin et al. 2019 ; Navratil et al. 2020; Marin et al. 2021; Kim et al. 2022; Cascante et al. 2023), it is likely that true levels of dispersal are substantially greater than 0. Further research based on imaging and/or spatial genomic methods will be required to detail the spatial distribution of daughter cells after each mitotic division, improving understanding of the location of aneuploid cells within mosaic embryos and how these patterns change throughout development (McDole et al. 2018).

For simplicity, our model represents the ploidy of a given cell as a binary variable (aneuploid vs euploid) and does not consider the specific chromosome affected or the category of aneuploidy (gain vs loss of maternal vs paternal homologs). The main consequence of these simplifications is that we do not consider the possibility of "rescue" events where aneuploid lineages revert to euploidy due to subsequent mitotic errors. We believe that these simplifications are nevertheless justified to avoid model overparameterization, especially given current evidence that aneuploidy rescue is rare in blastocyst-stage human embryos (based on the paucity of uniparental disomy) (McCoy et al. 2023). Because gains and losses are not distinguished, we do not account for the possibility that sampling of cells possessing a reciprocal gain and loss (e.g. trisomy and monosomy chromosome 1) within a single biopsy could evade detection by bulk sequencing. Although we expect this phenomenon to remain rare due to the requirement for balanced representation of both lineages, it would nonetheless cause us to underestimate the true incidence of mitotic error and mosaicism in human blastocysts. Similarly, our model does not account for potential changes in rates of mitotic error across cell divisions, and indeed, the first 2 cell divisions are known to be particularly error-prone (Currie et al. 2022). In the work presented here, we estimated a single fixed probability of mitotic error across all cell divisions that produced an embryo. It is likely that this fixed estimate is an underestimate of the rate of error for initial divisions but an overestimate of the rate of error for later divisions. However, it is worth noting that early divisions may have a greater influence on our estimates, given their greater impact on observed patterns of mosaicism. Allowing for varying rates of mitotic error across rounds of cell division is an interesting future direction but may again introduce too many additional parameters given the dimensionality of the target data.

Related to the above considerations, we emphasize that published PGT-A data are subject to several selection and ascertainment biases. One of the most important such biases is that only embryos that survive to the blastocyst stage and exhibit good morphology are typically biopsied and reported in retrospective clinical studies (Forman et al. 2013; Viotti, Victor et al. 2021; ESHRE Working Group on Chromosomal Mosaicism et al. 2022). Meanwhile, the approximately half of embryos that arrest prior to blastocyst formation are highly enriched for abnormal cell divisions and complex forms of chromosomal mosaicism, as well as meiotic aneuploidies to a lesser extent (Chatzimeletiou et al. 2005; Santos et al. 2010; Maurer et al. 2015; McCoy et al. 2023). Additional biases include characteristics of the patient population, which could be enriched for particular forms of aneuploidy associated with infertility diagnosis. However, our previous research showed that common clinical indications for PGT-A exhibit only modest associations with meiotic or mitotic aneuploidy (McCoy et al. 2015), suggesting that this clinical cohort bias may

be negligible. Nevertheless, we emphasize that our inferred rates of both meiotic and mitotic error should be interpreted as estimates for the clinically tested samples and may not generalize to samples of in vivo conceived embryos from the broader human population.

Our research findings underscore the distinction between evidence of mosaicism within a biopsy, the existence of mosaicism within an embryo, and the clinical implications of mosaicism for pregnancy outcomes. While meiotic aneuploidies are generally harmful for development, mosaic aneuploidies that persist to the blastocyst stage are potentially compatible with healthy birth (Greco et al. 2015; Victor, Tyndall et al. 2019). This hypothesis was recently supported by a nonselection clinical trial which demonstrated that embryos exhibiting mosaicism as assessed by PGT-A had pregnancy outcomes equivalent to euploid embryos (Capalbo et al. 2021), though this may depend on the specific definition of mosaicism (Viotti, McCoy et al. 2021). Our findings also support this hypothesis, as they suggest that most embryos yielding euploid biopsies are low-level mosaics. While numerous forms of pathogenic mosaic aneuploidy have been reported (Biesecker and Spinner 2013), current evidence suggests that mosaic aneuploidies identified in blastocyst-stage embryos are rarely detected at later stages of pregnancy or at birth (Viotti, Victor et al. 2021) (although see Greco et al. 2023). This observation is consistent with a model of selection against aneuploid cells within mosaic embryos during peri- and postimplantation development, recently supported by data from mouse and human embryo models (Bolton et al. 2016; Singla et al. 2020; Yang et al. 2021), as well as analyses of single-cell genomic data from human embryos (Starostik et al. 2020; Yang et al. 2021). Future extensions of our model may consider such mechanisms of intraembryo negative selection, especially if supplemented by data from additional embryonic and fetal time points.

Together, our work helps expand beyond the limited snapshot of embryo biopsies by estimating the fundamental rates of cell division errors that best explain clinical data. Understanding these basic error mechanisms, the evolutionary forces that shape them, and their consequences for development enhance basic understanding of human development, while building the foundations for future clinical innovations.

## Methods

For each sampled combination of meiotic and mitotic error rates, we simulated a single biopsy from each embryo using the Tessera R package (Skinner et al. 2024). We then applied ABC using the R package EasyABC (Jabot et al. 2013) to infer posterior distributions of the meiotic and mitotic error rates by comparing the simulated biopsy data to published clinical data (Capalbo et al. 2021). Our approach is summarized in Fig. 1.

### Embryo model abstraction

Meiotic errors were simulated as producing embryos with entirely aneuploid cells. Mitotic errors were simulated as a given division of a euploid cell producing 2 aneuploid daughter cells. Upon selecting a meiotic and mitotic error rate, we simulated 8 rounds of mitotic cell division for a total of 256 cells. We repeated each simulation trial across 3 fixed dispersal levels: 0 (complete clustering of aneuploid cells), 0.5 (intermediate clustering of aneuploid cells), and 1 (maximum spatial separation of aneuploid cells). For the purpose of abstraction, aneuploidy was defined as a binary variable, ignoring the identity of individual chromosomes (see Discussion).

## Biopsy data reference

The clinical data used as the target for ABC are derived from Capalbo et al. (2021) and include 6,766 clinical trophectoderm biopsies performed in 2020 and 2021. Of these biopsies, 23.2% were euploid, 18.7% were mosaic, and 58.1% were aneuploid. Embryos in that study were classified as euploid if, based on estimates from copy number analysis, fewer than 20% of the biopsied cells were aneuploid, as mosaic if 20 to 70% of the cells were aneuploid, and as aneuploid if more than 70% of the cells were aneuploid for any chromosome (Capalbo et al. 2021).

## ABC algorithm

ABC involves simulating a large number of trials under a given model, comparing the simulation results to a set of target summary statistics, and identifying the parameters that best explain those data (Jabot et al. 2013). ABC is useful in situations such as these, where explicit likelihoods are not available but a parameterized simulator is available. Specifically, we used the adaptive population Monte Carlo ABC method from Lenormand et al. (2013) to automatically select a series of tolerance levels and stopping criteria. This method was demonstrated by the authors to produce high-quality estimates of the posterior distributions with fewer simulations.

We implemented this procedure with the ABC_sequential function and the methods="Lenormand" argument in the EasyABC package in R. Meiotic and mitotic error rates were drawn from uniform distributions (ranging from 0 to 1, exclusive), and each selected error rate pair was used to simulate 1,000 embryos (Jabot et al. 2013). The embryos and their biopsy summaries were then compared with the published clinical PGT-A results (Capalbo et al. 2021), and the posterior distributions were obtained.

## Posterior predictive distributions of embryos

To understand the characteristics of embryos given our inferences, we simulated 1,000,000 unique embryos by drawing meiotic and mitotic error rates from our posterior distributions (1,000 times), repeating this procedure for each of the 3 fixed dispersal levels. Proportions of fully euploid, fully aneuploid, and mosaic embryos were then tabulated for each dispersal level, as was the distribution of the proportion of aneuploid cells within mosaic embryos.

To understand the concordance between multiple biopsies of the same embryo, we simulated 2 consecutive biopsies from each of the above embryos using Tessera's biopsy functions. The first biopsy used the default index cell, while the second biopsy randomly selected one of the 256 cells as the index cell. Biopsy classification followed the thresholds from Capalbo et al. (2021), with copy number estimates fewer than 20% being euploid, between 20 and 70% being mosaic, and over 70% being aneuploid.

## Misclassification simulation

To investigate a scenario in which some biopsies classified as mosaic were in fact technical false positives, we reclassified mosaic biopsies from the Capalbo et al. (2021) study as euploid and aneuploid (from 0 to 100%, incrementing in steps of 10%), with misclassified embryos evenly reassigned to the euploid and aneuploid categories. Using these adjusted values as the new target, we repeated our ABC and posterior predictive sampling procedures described above across each of the 3 fixed dispersal levels.

## Data availability

All scripts to execute the simulations described in the study are available at https://github.com/mccoy-lab/aneuploidyRates. Clinical data used for parameter inference were published in Capalbo et al. (2021) (see their Figure A), Clarke et al. (2023) (see their Table 3), Rodrigo et al. (2020) (see their Table 1), and Munne et al. (2017) (see their Table 3). Supplemental material available at GENETICS online.

## Acknowledgments

The authors thank members of the McCoy Lab and Origins of Aneuploidy Research Consortium for helpful feedback and discussions. The authors also thank staff at the Advanced Research Computing at Hopkins (ARCH) core facility.

## Funding

This work was supported by a Johns Hopkins University Provost's Undergraduate Research Award to Q.Y., a National Science Foundation Graduate Research Fellowship (1746891) to S.A.C., and a US National Institutes of Health/National Institute of General Medical Sciences (NIH/NIGMS) Award R35GM133747 to R.C.M. ARCH was supported by the National Science Foundation (OAC1920103).

## Conflicts of interest

R.C.M. is a coinventor of a method for analysis of PGT-A data which is the subject of a US patent (12,322,509) assigned to Johns Hopkins University.

## Author contributions

Q.Y. was involved in software, validation, formal analysis, investigation, data curation, writing—original draft, writing—review and editing, and visualization. S.A.C. was involved in methodology, software, investigation, data curation, writing—original draft, writing—review and editing, supervision, project administration, and funding acquisition. M.I. was involved in investigation and writing—review and editing. R.C.M. was involved in conceptualization, methodology, resources, data curation, writing—review and editing, visualization, supervision, project administration, and funding acquisition.

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

*Editor: A. MacQueen*