## [Peer Review File · Genetics]

Approximate Bayesian computation supports a high incidence of chromosomal mosaicism in blastocyst-stage human embryos

Angela Yang, Sara Carioscia, Matthew Isada, and Rajiv McCoy

NOTE: The reviews and decision letters are unedited and appear as submitted by the reviewers.

In extremely rare instances and as determined by a Senior Editor or the EIC, portions of a review may be redacted. If a review is signed, the reviewer has agreed to no longer remain anonymous.

The review history appears in chronological order.

Review Timeline:

Submission Date:	2024-12-16
Editorial Decision:	2025-03-20
Resubmission Received:	2025-06-02
Accepted:	2025-06-28

March 19, 2025

GENETICS-2024-307729

Approximate Bayesian computation supports a high incidence of chromosomal mosaicism in blastocyst-stage human embryos

Dear Dr. McCoy:

Two experts in the field have reviewed your manuscript, and I have read it as well. I am pleased to inform you that, with minor revisions, it is potentially suitable for publication in GENETICS. The reviewers have comments and concerns that need to be addressed in a revised manuscript. You can read their reviews at the end of this email.

Most of the requests from the reviewers are for clarification or expanded discussion. However, it is most important that these are addressed, in most cases (as recommended) by amendment to the manuscript.

We look forward to receiving your revised manuscript. Please let the editorial office know approximately how long you expect to need for revisions.

Upon resubmission, please include:

1. A clean version of your manuscript;
2. A marked version of your manuscript in which you highlight significant revisions carried out in response to the major points raised by the editor/reviewers (track changes is acceptable if preferred);
3. A detailed response to the editor's/reviewers' comments and to the concerns listed above. Please reference line numbers in this response to aid the editors.

Additionally, please ensure that your resubmission is formatted for GENETICS.

<https://academic.oup.com/genetics/pages/general-instructions>

Follow this link to submit the revised manuscript: Link Not Available

Sincerely,

Paul Scheet
Associate Editor
GENETICS

Approved by:
Amy MacQueen
Senior Editor
GENETICS

Reviewer #1 :

Enclosed is a review of Yang et al's manuscript: "Approximate Bayesian computation supports a high incidence of chromosomal mosaicism in blastocyst-stage human embryos." In this manuscript, the authors explore through simulations, based on clinically-derived data, the incidence of aneuploid, euploid, and mosaic embryos and how different meiotic/mitotic rates affect these incidences. The authors use ABC to generate simulations and estimate probabilities of meiotic error per meiosis, mitotic error rates, and incidence of euploids. Overall, the manuscript is clear and well-written. I appreciate how easy to read the Results section is. In total, my comments are minor, mostly focusing on organization of the paper and additional details.

Below are my specific comments:

1. The first two parameters in the simulation (meiotic error rate and mitotic error rate) are clear, but I was a little confused about the rate of dispersal parameter. I would advise the authors to add some intuition about what this parameter entails.
2. Figure 4 is difficult to read - mainly because the histograms have to support the peak at cell proportion 100%. The difference in the distribution at the left tail is difficult to see, and this is important as the authors make a claim about the difference in skewness.
3. The clinical implications of these results should be mentioned, in short, in the abstract.

4. This is probably my biggest question about the methods: the authors describe ABC well, but I wonder if further justification of ABC in the methods section, mainly in comparison to other methods for simulation.
5. Are there other datasets that could provide the ABC with a different set of priors? The authors mention that these results are tailored to clinically-relevant parameters, but how heterogeneous are these inputs?
6. If the Viotti data is accessible in a repository, please link these in the availability statement. Code is also sparsely commented - it would be good to have some more guidance for users.

Reviewer #2 :

Review of the manuscript: "Approximate Bayesian computation supports a high incidence of chromosomal mosaicism in blastocyst stage human embryos"

Summary:

The authors present a method for inferring the rate of aneuploidy errors in meiosis and in each cell division (mitosis) in early embryos. The method is based on matching empirical data for the proportion of euploid, aneuploid, and mosaic embryos (as observed in preimplantation genetic testing) to the proportions generated by a model of embryo development. The results show mitotic error rates around 1-3%, implying that almost all blastocyst-stage embryos contain some number of aneuploid cells. These results are robust to some model assumptions.

Evaluation:

The method is elegant, and it successfully uses available data and simulation tools to address a key question in embryo development. The mitotic error rate during embryo development is not only biologically interesting, but it also has clinical implications for the evaluation of the live birth potential of human IVF embryos. The robustness of the results is quite surprising given the very small amount of empirical data used for fitting the model (and the uninformative prior).

Comments:

Major comments:

- It would be nice if the authors could somewhat expand the Discussion to talk about the assumptions that go into the simulator, because these details may affect the inferred error rates. Ideally, experimenting with additional simulation parameters could enhance the robustness of the conclusions. But I agree with the authors that the space of possible parameter combinations will rapidly explode and that the added insights may quickly reach diminishing returns.
- As I wrote above, the results are surprisingly robust given that the entire input data is only two data points (three proportions). I'll leave the following as optional, as I would not like to impose additional work. But I'm wondering whether some additional experimental data could be analyzed. For example, what are the corresponding proportions of each embryo type at specific days post-fertilization? Is there information on the proportion of mosaic cells in the biopsy? In other words, could the mosaic embryo type be split into low/medium/high mosaicism (or a corresponding quantitative measure)? A breakdown by maternal age would also be important, given that it strongly affects the aneuploidy rate, and it is unclear whether the mitotic error rate is affected as well. A breakdown by the indication for infertility, or the number of aneuploid chromosomes in the biopsy, would be nice as well. But, most importantly, I'm concerned about replication. What would be the proportion of embryo types in other labs? Are the results lab-specific or can be generalized? Given that all that's required is three proportions, such data may be not too difficult to obtain. As I wrote above, I leave this as optional.

Minor comments

- The authors mention a couple of times that the error rate is higher in the first two cell divisions. However, an error in chromosome segregation in the very first cell division should (theoretically) lead to an entirely aneuploid embryo. Is that correct? If this is the case (and this is how the simulator is implemented), then it is impossible to tease apart the aneuploidy error rate and that of the first division.
- Relatedly, in line 249, "The practical effect of this simplification is that we likely underestimate the rate of error for initial divisions". I think this is not obviously the conclusion. This is because each division creates a different proportion of aneuploid cells in the blastocyst, depending on the timing of the division. Thus, the first two divisions have an outsized impact on the probability to detect a mosaic embryo, so perhaps the inferred error rate is in reality closer to the error rates in the first divisions.
- In the Methods (e.g., line 293), the authors mention 8 mitotic cell divisions. However, this is really 8 "rounds" of cell division, and the number of mitoses should be 255. Is that right? Does the simulator indeed provide 255 opportunities for a mitotic error?
- In the simulations, what was the procedure used to declare that an embryo is mosaic? Was there a minimal or a maximal number of aneuploid cells for a mosaic classification?

Typos, phrasing, etc (no need to respond):

- Line 146: "We next used the ABC results to examine the set of simulated embryos that produce the biopsy results that best match Viotti et al". However, the rest of the paragraph suggests that it was not the best set of simulated embryos, but rather that embryos were simulated for this analysis based on sampling error rate parameters from the posterior distributions.
- Next, line 147: "To this end, for each dispersal level, we generated embryos from the posterior distributions..." It is a bit

misleading, because the simulation itself did not change with dispersal rates (if I understand correctly). It's just that the distribution of inferred error rates was different for each dispersal rate.

- In Figure 1, part (2), I think the figure would be clearer if the prior density of the error rate would be non-zero only in the range $[0, 1]$. (The current plot seems as if the density continues to infinity).
- Figure 3: The colors of the dots and the colorbar are not clear. What is the "Distance"?
- Lines 183-184: In the simulations of mis-classification, was each mosaic embryo converted into euploid or aneuploid with equal probability? The probabilities are not mentioned.
- Figure 5 caption: Line 198 - I think it should be x-axis instead of y-axis.
- Line 233: I think "dispersal=0" should be swapped with "dispersal=1".

We are grateful for the careful evaluation and constructive feedback from the reviewers and editor, which have helped to improve our work. Please see the original comments below in black, interspersed with our responses in blue.

Reviewer #1 :

Enclosed is a review of Yang et al's manuscript: "Approximate Bayesian computation supports a high incidence of chromosomal mosaicism in blastocyst-stage human embryos." In this manuscript, the authors explore through simulations, based on clinically-derived data, the incidence of aneuploid, euploid, and mosaic embryos and how different meiotic/mitotic rates affect these incidences. The authors use ABC to generate simulations and estimate probabilities of meiotic error per meiosis, mitotic error rates, and incidence of euploids. Overall, the manuscript is clear and well-written. I appreciate how easy to read the Results section is. In total, my comments are minor, mostly focusing on organization of the paper and additional details.

Thank you for your encouraging feedback and suggestions for improving the organization of our manuscript.

Below are my specific comments:

1. The first two parameters in the simulation (meiotic error rate and mitotic error rate) are clear, but I was a little confused about the rate of dispersal parameter. I would advise the authors to add some intuition about what this parameter entails.

We added text to the beginning of the Results where the concept of dispersal is introduced (lines 112-115) that provides readers with intuition for how to interpret low, medium, and high rates of dispersal.

2. Figure 4 is difficult to read - mainly because the histograms have to support the peak at cell proportion 100%. The difference in the distribution at the left tail is difficult to see, and this is important as the authors make a claim about the difference in skewness.

We reformatted this figure with a "broken" y-axis to better display the small proportions on the left side of the histograms as well as the large proportions on the right side of the histograms. The skewness and multimodality are now apparent—a reflection of the large number of aneuploid cells that can originate from chromosome mis-segregation events that affect early embryonic divisions.

3. The clinical implications of these results should be mentioned, in short, in the abstract.

We now briefly note the clinical implications of our results in the abstract (lines 35-38) and expand on this point in the Discussion (lines 339-358). As academic researchers in the basic sciences, we are careful not to over-state these implications, as the primary goal of our research is to use clinical data to inform biological understanding. Nevertheless, we agree that our results may offer a foundation for clinical interpretation, which we explain in the context of emerging clinical research.

4. This is probably my biggest question about the methods: the authors describe ABC well, but I wonder if further justification of ABC in the methods section, mainly in comparison to other methods for simulation.

This is a good suggestion. We note, however, that while ABC uses simulations, it is not itself a method for simulation. Conceptually, we think it is most relevant to contrast ABC with the derivation of an explicit likelihood function. To address this comment, we added a sentence to the Methods section “ABC algorithm” further justifying the application of ABC (lines 392-393; also see lines 78-86). ABC is uniquely suited for inferring unknown parameters in scenarios where simulation is possible given a set of parameters, but where an explicit expression of the likelihood function is unavailable, as is the case here. This complexity emerges from the spatial structure of the embryo and was implemented in the Tessera simulation package.

5. Are there other datasets that could provide the ABC with a different set of priors? The authors mention that these results are tailored to clinically-relevant parameters, but how heterogeneous are these inputs?

Thank you for this suggestion. The reported incidences of aneuploid, euploid, and mosaic biopsies (i.e., the inputs) indeed vary across studies, both due to characteristics of the patient population (e.g., maternal age) and due to technical differences in clinical practices or reporting.

While seeking to extend our analysis to additional input datasets, we realized that one such reporting discrepancy regards consideration of segmental (i.e., sub-chromosomal) aneuploidies. The input data used in our initial submission (Viotti et al., 2021) grouped whole-chromosome and segmental aneuploidies together. As such, the rates of meiotic and mitotic error that we previously inferred should be understood to reflect combined rates of both whole and segmental chromosome mis-segregation. Combining these categories is potentially misleading given the distinct originating mechanisms. We therefore decided to focus our analysis solely on whole-chromosome aneuploidies and the underlying rates of whole-chromosome mis-segregation.

To this end, we replaced all of our primary analyses with results generated using input PGT-A data from Capalbo et al. (2021) that solely reports incidences of whole-chromosome aneuploidies. To address the original comment of the reviewer, we also extended the analysis to three additional input PGT-A datasets that similarly report incidences of whole-chromosome aneuploidies (Clarke et al. 2023, Munne et al. 2017, and Rodrigo et al. 2020). These results are reported in the new section titled “Inferences are robust to variation across PGT-A datasets” and demonstrate that our findings are qualitatively consistent across a range of clinical biopsy results.

6. If the Viotti data is accessible in a repository, please link these in the availability statement. Code is also sparsely commented - it would be good to have some more guidance for users.

Embryo-level data are not used in our analysis—only the summarized proportions of euploid, mosaic, and aneuploid biopsies which are used as the target for ABC. We obtained these directly from the text of the cited papers. We updated the “Data and code availability” section to explicitly state where in each paper (i.e., Figures or Tables) these proportions are reported.

Thank you also for the suggestion about improving the readability of our code. We have re-organized the GitHub repository with READMEs and have expanded our comments within relevant scripts to guide readers/users.

Reviewer #2 :

Review of the manuscript: "Approximate Bayesian computation supports a high incidence of chromosomal mosaicism in blastocyst stage human embryos"

Summary:

The authors present a method for inferring the rate of aneuploidy errors in meiosis and in each cell division (mitosis) in early embryos. The method is based on matching empirical data for the proportion of euploid, aneuploid, and mosaic embryos (as observed in preimplantation genetic testing) to the proportions generated by a model of embryo development. The results show mitotic error rates around 1-3%, implying that almost all blastocyst-stage embryos contain some number of aneuploid cells. These results are robust to some model assumptions.

Evaluation:

The method is elegant, and it successfully uses available data and simulation tools to address a key question in embryo development. The mitotic error rate during embryo development is not only biologically interesting, but it also has clinical

implications for the evaluation of the live birth potential of human IVF embryos. The robustness of the results is quite surprising given the very small amount of empirical data used for fitting the model (and the uninformative prior).

Thank you very much for these detailed suggestions, which we used to improve our paper.

Comments:

Major comments:

- It would be nice if the authors could somewhat expand the Discussion to talk about the assumptions that go into the simulator, because these details may affect the inferred error rates. Ideally, experimenting with additional simulation parameters could enhance the robustness of the conclusions. But I agree with the authors that the space of possible parameter combinations will rapidly explode and that the added insights may quickly reach diminishing returns.

This is an excellent point. In the Discussion, we highlight several key assumptions of our model, including regarding the spatial distribution of aneuploid cells within mosaic embryos, intra-embryo selection within mosaic embryos (whereby euploid cells might possess fitness advantages compared to aneuploid cells in certain contexts), and rescue of aneuploidy (correction of aneuploid lineages by mitotic error in subsequent divisions). We provide conceptual arguments and references to support these modeling decisions. For example, in the case of spatial distribution, we vary the dispersal parameter over a range to evaluate the impact on our results. In the case of aneuploidy rescue, we disregard the phenomenon because it would introduce substantial model complexity (risking overfitting and computational intractability) and current data suggest it is very rare based on the low incidence of uniparental disomy.

In response to your comment, we added justification for two additional simplifications to the Discussion: 1) the decision to consider aneuploidy status as a binary variable rather than tracking copy number of all 23 pairs of chromosomes and 2) the related decision to disregard the possibility of reciprocal aneuploidies, where balanced representation of gain/loss of the same chromosome within a biopsy may go undetected.

- As I wrote above, the results are surprisingly robust given that the entire input data is only two data points (three proportions). I'll leave the following as optional, as I would not like to impose additional work. But I'm wondering whether some additional experimental data could be analyzed. For example, what are the corresponding proportions of each embryo type at specific days post-fertilization? Is there information on the proportion of mosaic cells in the biopsy? In other words, could the mosaic embryo type be split into low/medium/high mosaicism (or a corresponding quantitative measure)? A breakdown by maternal age would also be important, given

that it strongly affects the aneuploidy rate, and it is unclear whether the mitotic error rate is affected as well. A breakdown by the indication for infertility, or the number of aneuploid chromosomes in the biopsy, would be nice as well. But, most importantly, I'm concerned about replication. What would be the proportion of embryo types in other labs? Are the results lab-specific or can be generalized? Given that all that's required is three proportions, such data may be not too difficult to obtain. As I wrote above, I leave this as optional.

These are very interesting suggestions for extensions of our modeling approach. The relationship between the proportion of aneuploid cells within a TE biopsy and the proportion of aneuploid cells within an embryo was extensively considered by Skinner et al., so we did not focus on it in our own work. Nevertheless, we agree that in the future it may be possible to expand our analysis to additional embryonic timepoints (e.g., cleavage-stage blastomere biopsies) or other relevant axes of variation.

For our revisions, we focused on your main concern of replication of our results across varying input datasets. As noted above, such input data can vary due to random chance, features of the patient population (e.g., the maternal age distribution or infertility diagnoses), and technical variability across clinics or genetic testing providers. In doing so, we realized that one such reporting discrepancy regards consideration of segmental (i.e., sub-chromosomal) aneuploidies. The input data used in our initial submission (Viotti et al., 2021) grouped whole-chromosome and segmental aneuploidies together. As such, the rates of meiotic and mitotic error that we previously inferred should be understood to reflect combined rates of both whole and segmental chromosome mis-segregation. Combining these categories is potentially misleading given the distinct originating mechanisms. We therefore decided to focus our analysis solely on whole-chromosome aneuploidies and the underlying rates of whole-chromosome mis-segregation.

To this end, we replaced all of our primary analyses with results generated using input PGT-A data from Capalbo et al. (2021) that solely reports incidences of whole-chromosome aneuploidies. To address the original comment of the reviewer, we also extended the analysis to three additional input PGT-A datasets that similarly report incidences of whole-chromosome aneuploidies (Clarke et al. 2023, Munne et al. 2017, and Rodrigo et al. 2020). These results are reported in the new section titled "Inferences are robust to variation across PGT-A datasets" and demonstrate that our findings are qualitatively consistent across a range of clinical biopsy results.

Minor comments

- The authors mention a couple of times that the error rate is higher in the first two cell divisions. However, an error in chromosome segregation in the very first cell division should (theoretically) lead to an entirely aneuploid embryo. Is that correct? If this is the case (and this is how the simulator is implemented), then it is impossible to tease apart the aneuploidy error rate and that of the first division.

You are correct that a chromosome segregation error in the first mitotic division leads to an entirely aneuploid embryo. Our model assigns both daughter cells as aneuploid and does not distinguish between chromosome gain and loss. You are therefore correct that even with a complete view of all embryonic cells, it would be impossible to determine whether an entirely aneuploid embryo originated via meiotic or mitotic error. The challenge is even more pronounced in the case of embryo biopsies (i.e., an incomplete, spatially restricted sample of a given embryo), where the same pattern can arise from a variety of histories of faithful or erroneous cell divisions that produced an embryo. For example, a euploid embryo biopsy can originate from an entirely euploid embryo or an embryo with low or even high level mosaicism. Indeed, this is the very challenge that motivates our research.

Nevertheless, the fact that the different mechanisms of error (meiotic error, early mitotic error, late mitotic error) produce different distributions of outcomes enables inference of the unknown error rate parameters. Our results show that the rates of meiotic and mitotic error are identifiable, especially when dispersal is non-zero (**Fig. 2 & Fig. 3**). In summary, even though the same mechanism can occasionally produce identical observed outcomes (at the level of an embryo biopsy or even an entire embryo), the difference in the distribution of outcomes that result from each mechanism allows the rates of each mechanism to be inferred based on a large sample of embryo biopsies.

- Relatedly, in line 249, "The practical effect of this simplification is that we likely underestimate the rate of error for initial divisions". I think this is not obviously the conclusion. This is because each division creates a different proportion of aneuploid cells in the blastocyst, depending on the timing of the division. Thus, the first two divisions have an outsized impact on the probability to detect a mosaic embryo, so perhaps the inferred error rate is in reality closer to the error rates in the first divisions.

Thank you for pointing this out. We agree that the text was unclear as previously stated. You are correct that errors in earlier cell divisions have an outsized impact on the embryo and are more likely to be detected within biopsies, given the propagation of aneuploidies to descendant cells. In a given simulation, a single mitotic error probability is used to simulate 255 cell divisions, per your next comment. Because it is known that the first two cell divisions are more prone to chromosome mis-segregation than later cell divisions, we expect that the mitotic error probability of early cell divisions is somewhat higher than our inferred rate. This point holds even given your point that our inferred rate could be closer to the error rates in initial divisions than later divisions, as the former have a greater impact on the observed biopsy outcomes. We have updated the text in the Discussion (lines 313-320) to explain this point.

- In the Methods (e.g., line 293), the authors mention 8 mitotic cell divisions. However, this is really 8 "rounds" of cell division, and the number of mitoses should

be 255. Is that right? Does the simulator indeed provide 255 opportunities for a mitotic error?

You are correct; we model 8 rounds of cell divisions with 255 mitoses. We added the word “rounds” to the original sentence for clarification (line 376).

- In the simulations, what was the procedure used to declare that an embryo is mosaic? Was there a minimal or a maximal number of aneuploid cells for a mosaic classification?

Thank you for pointing out this omission from the text. In the simulated biopsies, we consider a biopsy mosaic if any but not all of the 5 sampled cells is aneuploid. This is consistent with cell line mixing experiments showing that low-coverage whole-genome sequencing is sensitive for detecting copy number displacement $\geq 20\%$ of diploid expectations (e.g., see <https://pubmed.ncbi.nlm.nih.gov/28393271/>). In the simulated embryos, we consider an embryo mosaic if any cell in the embryo is aneuploid. We added text to the Results section “ABC enables inference of rates of meiotic and mitotic error” (lines 120-121) describing our criteria for categorizing biopsies and to the Results section “Evidence that few embryos are fully euploid” (lines 187-189) describing our criteria for categorizing embryos.

Typos, phrasing, etc (no need to respond):

Thank you for these detailed comments and corrections! We have addressed all of these issues in their respective locations in the revised text.

- Line 146: "We next used the ABC results to examine the set of simulated embryos that produce the biopsy results that best match Viotti et al". However, the rest of the paragraph suggests that it was not the best set of simulated embryos, but rather that embryos were simulated for this analysis based on sampling error rate parameters from the posterior distributions.
- Next, line 147: "To this end, for each dispersal level, we generated embryos from the posterior distributions..." It is a bit misleading, because the simulation itself did not change with dispersal rates (if I understand correctly). It's just that the distribution of inferred error rates was different for each dispersal rate.
- In Figure 1, part (2), I think the figure would be clearer if the prior density of the error rate would be non-zero only in the range [0,1]. (The current plot seems as if the density continues to infinity).
- Figure 3: The colors of the dots and the colorbar are not clear. What is the "Distance"?
- Lines 183-184: In the simulations of mis-classification, was each mosaic embryo converted into euploid or aneuploid with equal probability? The probabilities are not mentioned.

-
- Figure 5 caption: Line 198 - I think it should be x-axis instead of y-axis.
 - Line 233: I think "dispersal=0" should be swapped with "dispersal=1".

June 28, 2025

RE: GENETICS-2025-308243

Dr. Rajiv C. McCoy
Johns Hopkins University
Biology
3400 N Charles St
Mudd 144
Baltimore, Maryland 21218

Dear Dr. McCoy:

Congratulations, your manuscript titled "Approximate Bayesian computation supports a high incidence of chromosomal mosaicism in blastocyst-stage human embryos" is accepted for publication in GENETICS! Many thanks for submitting your research to the journal.

Reviewer #2 has one suggestion for improving the manuscript that you may want to consider. You can view both reviewers' comments at the bottom of this email.

As part of our efforts to make titles of articles published in GENETICS more accessible to our broad readership, we often suggest different titles for accepted manuscripts. We offer these variants for your consideration though we'll use whatever title you include in the final version of your manuscript that you submit to the Editorial Office.

TITLE Suggestions:

Incidence of chromosomal mosaicism in human blastocyst embryos estimated by approximate Bayesian computation

Estimating rates of cell division errors causing human pregnancy loss with approximate Bayesian computation

Estimating incidence of chromosomal mosaicism in human blastocyst embryos by approximate Bayesian computation

To Proceed to Publication:

1. Format your article according to GENETICS style: <https://academic.oup.com/genetics/pages/general-instructions>

2. Ensure that you comply with data and community resource citation guidelines:
<https://academic.oup.com/genetics/pages/general-instructions#Data-Policy>

3. Upload your final files at <https://genetics.msubmit.net>

4. Add oupsupport@scipris.com and genetics.oup@novatechset.com (or the domains @scipris.com and @novatechset.com) to your email program's "safe senders" list. You will be contacted by both at various points during the production process.

Notes:

- Your currently-accepted manuscript (unedited, as submitted, reviewed, and accepted) will be published at GENETICS and deposited into PubMed as an Advance Access article. Notify sourcefiles@thegsajournals.org before signing your license if you do not wish to publish your article via Advance Access.

- We invite you to submit an original color figure related to your paper for consideration as cover art. Please email your submission to the editorial office or upload it with your final files. You can submit a small-sized image for evaluation, and if selected, the final image must be a TIFF file 2513px wide by 3263px high (8.375 by 10.875 inches; resolution of 600ppi). Please avoid graphs and small type.

- After files are sent to Oxford University Press we use SciPris to manage article licensing and payment. If you do not have a SciPris account, you will receive an email from no-reply@scipris.com to sign up to use Oxford University Press' author portal. After logging in, follow the online instructions to sign your license and arrange any payment due.

If you have any questions or encounter any problems while uploading your accepted manuscript files, please email the editorial office at sourcefiles@thegsajournals.org.

Sincerely,

Amy MacQueen
Senior Editor
GENETICS

Approved by:
Howard Lipshitz
Editor in Chief
GENETICS

Review comments (if applicable):

Reviewer #1 :

I thank the authors for adequately addressing my comments.

Reviewer #2 :

Thank you for addressing my comments and performing the necessary re-analyses. Congrats on a nice paper.

I have one question left. Could the authors please provide, from their simulations, the proportion of non-aneuploid embryos (i.e., fully euploid or mosaic) whose 5-cell biopsy is entirely aneuploid?

This is clinically a very important parameter. For example, a paper just came out reporting the live birth of healthy twins whose biopsy was entirely aneuploid. [https://www.fertstert.org/article/S0015-0282\(25\)00538-2/fulltext](https://www.fertstert.org/article/S0015-0282(25)00538-2/fulltext)

Knowing the expected proportion of such cases would be very useful. I believe this information should be very easy to extract from the simulations.

Shai Carmi